# Surgical site peptidylarginine deaminase 4 (PAD4), a biomarker of NETosis, correlates with insulin resistance in total joint arthroplasty patients: A preliminary report

Vitor F. Martins[1,2☯], Christopher R. Dobson[1☯], Maedha Begur[1], Jesal Parekh[1], Scott T. Ball[1], Francis Gonzalez[1], Jan M. Hughes-Austin[1], Simon Schenk[1,2]*

1 Department of Orthopaedic Surgery, University of California San Diego, La Jolla, CA, United States of America, 2 Biomedical Sciences Graduate Program, University of California San Diego, La Jolla, CA, United States of America

☯ These authors contributed equally to this work.
* sschenk@ucsd.edu

**Data Availability Statement:** All relevant data are within the paper and its Supporting Information files.

## Abstract

While obesity and insulin resistance are known risk factors for wound complications after total joint arthroplasty (TJA), the biologic causes remain to be elucidated. Recently, neutrophil extracellular trap formation (NETosis) was identified as a mediator of delayed wound healing in insulin resistant states. Herein, we explored the relationship between obesity, insulin resistance and biomarkers of NET formation in TJA subjects. We enrolled 14 obese (body mass index [BMI]$\geq$30 kg/m$^2$), and 15 lean (BMI<30 kg/m$^2$) subjects undergoing primary knee or hip TJA. On the day of surgery, skeletal muscle proximal to the operated joint and plasma were collected. Protein abundance of NETosis biomarkers, peptidylarginine deaminase 4 (PAD4) and neutrophil elastase (NE) were assessed in skeletal muscle by immunoblotting and metabolic parameters (glucose, insulin, triglycerides, free fatty acids) and cell-free double-stranded DNA (cf-dsDNA) were assessed in plasma and were correlated with obesity and insulin resistance (as measured by the homeostatic model assessment for insulin resistance). When comparing lean and obese subjects, there were no significant differences in plasma cf-dsDNA or skeletal muscle NE or PAD4 abundance. In contrast, skeletal muscle PAD4 abundance, but not NE or plasma cf-dsDNA, was positively correlated with insulin resistance. Compared to insulin sensitive subjects, insulin resistant TJA subjects have higher expression of PAD4 at the surgical site and therefore may have higher rates of NET formation, which may lead to delayed surgical site wound healing.

## Introduction

Total joint arthroplasty is an increasingly common surgery for orthopaedic surgeons, with an estimated prevalence of 7 million patients in the United States and 1 million total hip and knee arthroplasties performed each year [1, 2]. A potentially disastrous complication of total joint

**Funding:** This work was supported, in part, by U.S. National Institutes of Health grants, T32 AR060712 and F30 DK115035 to V.F.M. (https://www.nih. gov), Graduate Student Research Support from the UC San Diego Institute of Engineering in Medicine and the Office of Graduate Studies (https://iem. ucsd.edu) to V.F.M, a Resident Research Grant from the Orthopaedic Research and Education Foundation (https://www.oref.org) to C.R.D., and a Medical Student Training in Aging Research Grant (https://www.afar.org/grants/mstar) to M.B. The funders had no role in study design, data collection and analysis, decision to publish, or preparation of the manuscript.

**Competing interests:** The authors have declared that no competing interests exist.

arthroplasty is post-operative wound infection at the surgical site, which can lead to periprosthetic joint infection and the need for revision arthroplasty.

Disregarding iatrogenic causes, one of the major risk factors for periprosthetic joint infection is obesity [3–5]. Numerous studies have demonstrated a link between obesity and delayed wound healing [6–8], which may, in part, explain its contribution to increased risk of periprosthetic joint infection. The possible molecular mechanisms behind this include decreased capillary proliferation [8], decreased keratinocyte proliferation [9], and/or increased inflammation and oxidative stress [10]. However, the pathophysiology underlying how obesity leads to delayed wound healing remains elusive. Notably, type 2 diabetes and insulin resistance, which are closely tied to obesity [11], also delay wound healing [12–15]. In fact, recently type 2 diabetes and hyperglycemia were individually found to delay wound healing in mice via the priming of neutrophils to undergo neutrophil extracellular trap (NET) formation, in a process referred to as NETosis [14, 16].

NET formation was originally identified as a host defense mechanism in which neutrophils release decondensed chromatin and anti-pathogenic proteins to eliminate infectious organisms [17]. At a molecular level, NET formation occurs via the activation of peptidylarginine deaminase 4 (PAD4), a nuclear enzyme that promotes the citrullination and reduction of histones, weakening their interactions with DNA, thus facilitating the decondensation of chromatin [18–21]. The decondensed chromatin is combined with anti-pathogenic proteins such as neutrophil elastase (NE), before being released outside of the cell as cell-free double-stranded DNA (cf-dsDNA) to trap and destroy virulent factors [17, 22]. Interestingly, while NET formation is protective against infection due to the sequestering and destroying pathogens, it can be harmful due to causing delayed wound healing [14, 15]; thus, whether the presence of NETs is protective or harmful towards wound infection is still an open question in the field.

Since ~35% of the US population is classified as obese [23] and 90% of type 2 diabetes patients are obese [24], elucidating potential contributing factors to delayed wound healing in total joint arthroplasty patients is of upmost importance, as the likelihood of a surgeon operating on this population is high. Considering the interrelationship between NET formation in wounds and delayed wound healing in insulin resistant states [14, 16], understanding how these factors interact in total joint arthroplasty patients could help establish recommendations for pre- and post-operative treatment in surgical patients. Accordingly, in this study, our aim was to investigate the relationship between markers of NET formation and the degree of adiposity and insulin resistance in total joint arthroplasty patients. We hypothesized that obese and insulin resistant patients would have higher markers of NETosis in skeletal muscle within the surgical site.

## Materials and methods

### Subjects

This study received Institutional Review Board approval from the University of California San Diego Human Research Protections Program (IRB: 151506; Approved 10/22/2015) and written informed consent was obtained from each patient before the initiation of the study. From January–March 2016 we enrolled subjects who were undergoing primary total joint arthroplasty (hip or knee) in the Department of Orthopedic Surgery at the University of California, San Diego. Due to variations in the underlying pathophysiology of indications for total joint arthroplasty, only subjects with degenerative osteoarthritis as their indication for surgery were included in this study. Exclusion criteria included malignancy, inflammatory arthritis, infection, trauma, and/or revision arthroplasty. Subjects enrolled in this study were grouped into "Obese" (BMI $\geq$ 30 kg/m$^2$) or "Lean" (BMI < 30 kg/m$^2$). Subject characteristics were recorded from electronic medical records and are summarized in Table 1. After metabolic labs were

**Table 1. Subject characteristics and labs.**

| | Lean | | | Obese | | | p-value |
|---|---|---|---|---|---|---|---|
| No. of patients | | 15 | | | 14 | | — |
| Sex (F/M) | | 6/9 | | | 9/5 | | 0.272 |
| Age (yrs) | 65.3 | ± | 3.5 | 58.9 | ± | 4.1 | 0.245 |
| Height (m) | 1.72 | ± | 0.03 | 1.67 | ± | 0.06 | 0.440 |
| Weight (Kg) | 75.6 | ± | 3.0 | 99.0 | ± | 6.1 | **0.002** |
| BMI (Kg/m$^2$) | 25.5 | ± | 0.6 | 35.2 | ± | 1.3 | **<0.001** |
| Surgery (Hip/Knee) | | 12/3 | | | 8/6 | | 0.245 |
| Fasting Glucose (mg/dL) | 92.0 | ± | 3.5 | 96.6 | ± | 8.2 | 0.595 |
| Fasting Insulin (μIU/mL) | 3.29 | ± | 0.46 | 8.08 | ± | 1.68 | **0.009** |
| HOMA-IR | 0.77 | ± | 0.12 | 2.03 | ± | 0.46 | **0.010** |
| Hb A1C (%) | 5.21 | ± | 0.10 | 5.46 | ± | 0.56 | 0.242 |
| FFA (mM) | 0.69 | ± | 0.06 | 0.80 | ± | 0.08 | 0.263 |
| TG (mg/dL) | 78.5 | ± | 11.8 | 85.9 | ± | 8.7 | 0.624 |

F, Female; M, Male; BMI, Body Mass Index; HOMA-IR, homeostatic model assessment of insulin resistance; Hb A1C, hemoglobin A1C; FFA, free fatty acids; TG, triglycerides. Unpaired t-test or Fisher exact test (sex, surgery) obese versus lean. Bolded values indicate significance at p < 0.05. Data reported as mean ± SEM.

processed (Table 1), the 14 obese subjects were stratified into "Insulin Resistant" (IR; homeo-static model assessment for insulin resistance [HOMA-IR] >2) [25, 26] or "Insulin Sensitive" (IS; HOMA-IR <2) (Table 2). Notably, all lean subjects were insulin sensitive. HOMA-IR was calculated using fasting plasma glucose and fasting plasma insulin concentrations [25, 26]: HOMA-IR = Fasting Glucose (mg/dL)*Fasting Insulin (μIU/mL)/405.

## Sample collection

On the day of surgery, blood was collected pre-operatively (n = 29; subjects were fasted >10 h), before anesthesia, and skeletal muscle tissue around the incision site was collected during

**Table 2. Subject characteristics and labs stratified by insulin resistance.**

| | IS | | | IR | | | p-value |
|---|---|---|---|---|---|---|---|
| No. of patients | | 7 | | | 7 | | — |
| Sex (F/M) | | 4/3 | | | 5/2 | | 0.999 |
| Age (yrs) | 55.4 | ± | 7.0 | 62.4 | ± | 4.3 | 0.410 |
| Height (m) | 1.65 | ± | 0.11 | 1.69 | ± | 0.06 | 0.773 |
| Weight (Kg) | 92.7 | ± | 10.5 | 105.3 | ± | 6.3 | 0.330 |
| BMI (Kg/m$^2$) | 33.4 | ± | 1.5 | 37.1 | ± | 2.0 | 0.159 |
| Surgery (Hip/Knee) | | 2/5 | | | 4/3 | | 0.592 |
| Fasting Glucose (mg/dL) | 78.0 | ± | 7.5 | 115.3 | ± | 11.0 | **0.016** |
| Fasting Insulin (μIU/mL) | 3.64 | ± | 0.92 | 12.51 | ± | 2.20 | **0.003** |
| HOMA-IR | 0.69 | ± | 0.22 | 3.37 | ± | 0.51 | **<0.001** |
| Hb A1C (%) | 5.17 | ± | 0.19 | 5.75 | ± | 0.28 | 0.112 |
| FFA (mM) | 0.69 | ± | 0.10 | 0.91 | ± | 0.11 | 0.156 |
| TG (mg/dL) | 90.2 | ± | 11.5 | 81.5 | ± | 13.7 | 0.634 |

IS, insulin sensitive; IR, insulin resistant; F, Female; M, Male; BMI, Body Mass Index; HOMA-IR, homeostatic model assessment of insulin resistance; Hb A1C, hemoglobin A1C; FFA, free fatty acids; TG, triglycerides. Unpaired t-test or Fisher exact test (sex, surgery) IS versus IR. Bolded values indicate significance at p < 0.05. Data reported as mean ± SEM.

surgery, without ischemia. Venous blood samples were obtained in EDTA tubes and centrifuged at 1,600 *g* for 10 minutes and then 16,000 *g* for 5 minutes to obtain plasma samples. Plasma was subsequently frozen in liquid nitrogen and stored at –80˚C for future measurement of glucose, insulin, triglycerides (TG), free fatty acids (FFA), and cell-free double-stranded DNA (cf-dsDNA). Skeletal muscle tissue (gluteus medius for hip and vastus medialis oblique for knee) was obtained from the surgical site, rinsed in sterilized saline, blotted dry, and snap frozen in liquid nitrogen. Samples were stored at -80˚C for subsequent immunoblotting analysis.

## Plasma measurements

Plasma glucose concentration was measured by the glucose oxidase method (Thermo Fisher Scientific, Waltham, MA, USA). Plasma insulin concentration was measured using a commercially available kit (Human Insulin ELISA; 80- INSHUE01.1; ALPCO Diagnostics, Salem, NH, USA). Plasma TG concentration was measured using a commercially available kit (L-Type Triglyceride M; Wako Diagnostics, Richmond, VA). Plasma FFA concentration was measured by colorimetric assay (NEFA-HR(2) 995–34791; Wako Diagnostics). cf-DNA was extracted from plasma samples using QIAamp DNA Mini prep (Qiagen, Germantown, MD, USA). cf-dsDNA was quantified from extracted cf-DNA utilizing Quant-iT PicoGreen dsDNA kit (Invitrogen, Carlsbad, CA, USA), and compared to a standard curve using a Lambda DNA standard.

## Immunoblotting

Tissues were homogenized using an electronic overhead homogenizer (Caframo, Ontario, Canada) in ice-cold lysis buffer (20 mM Tris-HCl, pH 7.4, 150 mM NaCl, 1% NP-40, 20mM NaF, 2 mM EDTA, pH 8.0, 2.5 mM NaPP, 20 mM b-glycerophosphate, 10% glycerol) which contained appropriate protease inhibitors (Pefabloc SC and cOmplete protease inhibitor cocktail, MilliporeSigma Burlington, MA). Lysates were then rotated end over end for 2 hours at 4˚C and centrifuged (12,000 rpm, 20 min, 4˚C). The supernatant was then frozen in liquid nitrogen and stored at -80˚C for subsequent analysis. SDS-PAGE was performed using standard methods, as previously described [27]. Briefly, 20μg of total tissue protein extract was boiled in 1X Laemmeli sample buffer and loaded on to 12% Criterion XT Bis-Tris gels (Bio-Rad, Hercules, CA, USA). The following primary antibodies were used: peptidylarginine deiminase, type IV (PAD4; GTX113945) and neutrophil elastase (NE; GTX113175) from Gene-Tex (Irvine, CA, USA), and glyceraldehyde-3-phosphate dehydrogenase (GAPDH; 10R-G109A) from Fitzgerald Industries (Acton, MA, USA). Horseradish peroxidase-conjugated anti-mouse or anti-rabbit secondary antibodies were from Bio-Rad. The blots were developed using the Pierce ECL reagent from Thermo Fisher Scientific and quantified using Image Lab Software (Bio-Rad) within the linear range.

## Statistics

Minimum sample size was calculated as 14 per group and was performed using G*Power [28] utilizing effect sizes (power of 0.80 and α of 0.05) from previously published data for PAD4 [14], NE [16] and cf-dsDNA [16]. Statistical analyses were performed using Prim 8 (GraphPad Software Incorporated, La Jolla, CA, USA) and R (R Foundation for Statistical Computing, Vienna, Austria) [29]. Data were analyzed by fisher exact tests (for sex and surgery type), Student's two-sample *t* tests (age, height, weight, BMI, plasma parameters and NETosis markers), one-way ANOVA (NETosis markers in lean, obese IS, and obese IR subjects), followed by Sidak's multiple comparison test when appropriate. *F*-test and goodness-of-fit was used for simple (BMI vs NETosis markers, and insulin resistance vs NETosis markers) or multiple

linear regressions (BMI and Insulin resistance vs NETosis markers). Significant differences were set at $p < 0.05$. All data are expressed as means ± SEM.

## Results

### Subject characteristics and metabolic labs

Our cohort included 29 subjects, of which 15 were lean and 14 were obese (Table 1). There were no significant differences based on sex, age, height, or surgery type among the groups. When assessing the metabolic parameters of these subjects, the obese group had higher fasting insulin and HOMA-IR (Table 1). There were no significant differences in fasting plasma glucose, hemoglobin A1C, plasma FFA, or plasma TG concentrations.

### Subject characteristics–obese subjects stratified by insulin resistance

After metabolic labs were processed, obese subjects were stratified by insulin resistance based on calculated HOMA-IR levels [25, 26] (Insulin Sensitive [IS]: HOMA-IR <2; Insulin Resistant [IR]: HOMA-IR >2). When stratified by IR, there were no differences based on sex, age, height, weight, BMI, or surgery type. In assessing metabolic parameters of these two groups, obese IR subjects had significantly higher fasting glucose, fasting insulin and HOMA-IR compared to obese IS subjects. Obese IR subjects had comparable hemoglobin A1C, plasma FFA, and plasma TG concentrations to obese IS subjects (Table 2).

### cf-dsDNA in TJA patients

cf-dsDNA concentration was not different between lean and obese patients (Fig 1A). Furthermore, when obese patients were stratified by insulin resistance, cf-dsDNA concentration was

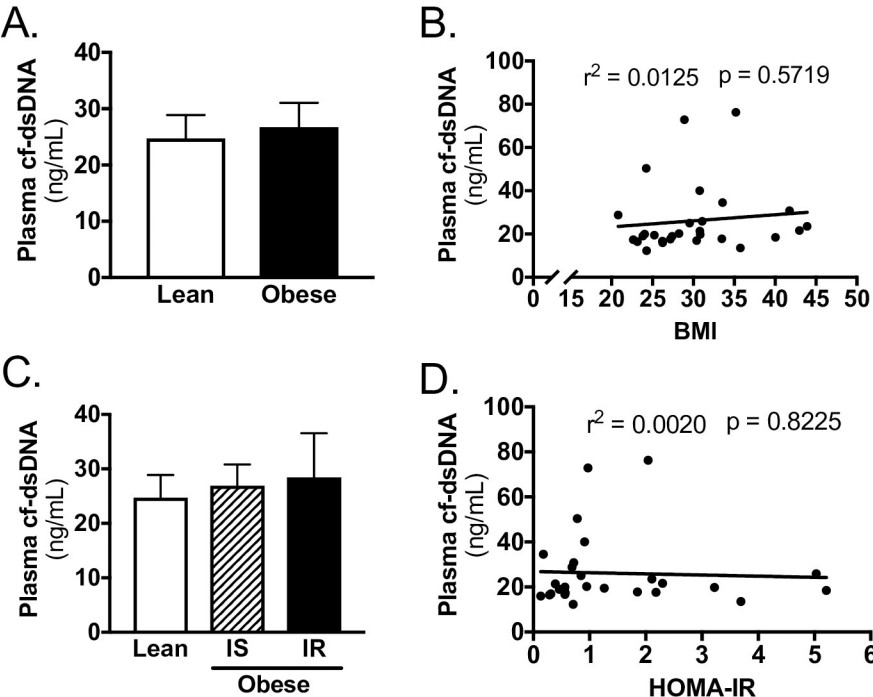

**Fig 1. cf-dsDNA does not correlate with obesity or insulin resistance in TJA subjects.** A) Plasma cf-dsDNA for lean and obese subjects. B) Linear regression of plasma cf-dsDNA with BMI of subjects. C) Plasma cf-dsDNA for lean, obese insulin sensitive (IS), and obese insulin resistant (IR) subjects. D) Linear regression of plasma cf-dsDNA with HOMA-IR of subjects. Data reported as mean ± SEM.

not different between subjects (Fig 1C). In regression analysis, there was no association between cf-dsDNA and BMI (Fig 1B, S1 Table) or HOMA-IR (Fig 1D, S1 Table).

### Skeletal muscle markers of NETosis—stratified by obesity

The protein abundance of NE and PAD4 in skeletal muscle from the surgical site were not different between lean and obese subjects (Fig 2A and 2B). In regression analysis, there was no association between NE or PAD4 and BMI (Fig 2C, 2D, S1 Table).

### Skeletal muscle markers of NETosis—stratified by insulin resistance

Obese IR subjects had significantly higher concentrations of PAD4, but not NE, in skeletal muscle from the surgical site, as compared to lean subjects (Fig 3A and 3B). In contrast, Obese IS subjects had comparable concentrations of PAD4 and NE to lean subjects (Fig 3A and 3B). In regression analysis, PAD4 abundance positively and significantly correlated with insulin resistance (Fig 3D, S1 Table), while NE did not (Fig 3C, S1 Table). Furthermore, when BMI and HOMA-IR were analyzed through multiple linear regression, PAD4 positively correlated with HOMA-IR, but not with BMI (S1 Table).

## Discussion

Perhaps the most feared complication following total joint arthroplasty is periprosthetic joint infection, which can lead to significant burden on both the treating physician and patient, as these infections often require multiple surgeries and are not always curative [30]. Obesity

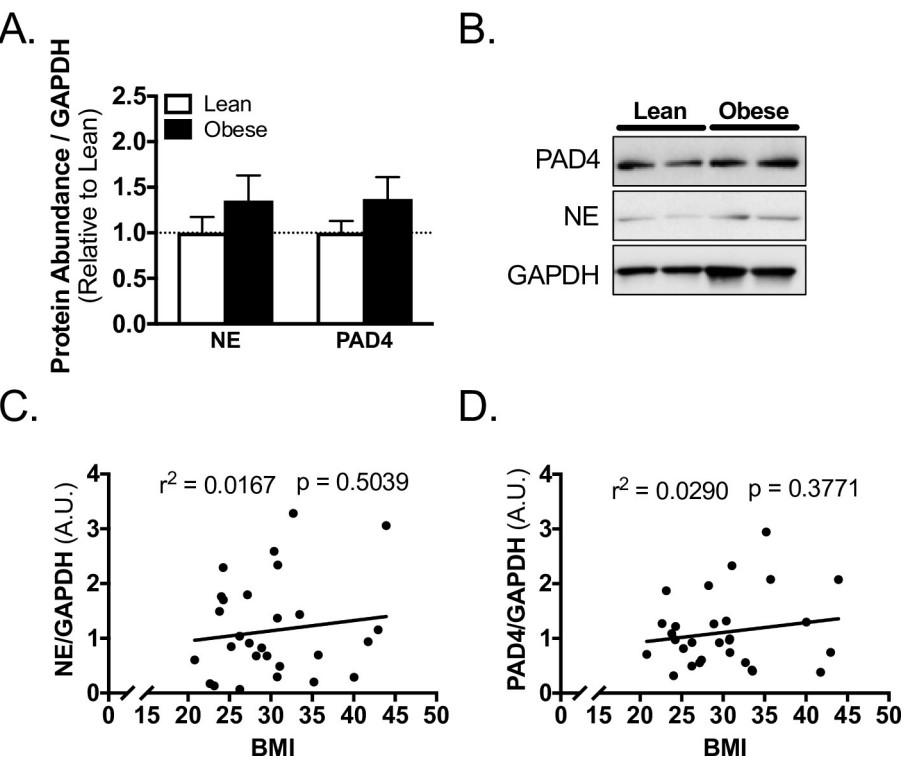

**Fig 2. Skeletal muscle biomarkers of NETosis do not correlate with obesity in TJA subjects.** A) Western blot quantification and B) representative blots for PAD4, NE, and loading control GAPDH abundance in skeletal muscle of lean and obese subjects. Linear regression of C) NE, and D) PAD4 with BMI of subjects. Data reported as mean ± SEM.

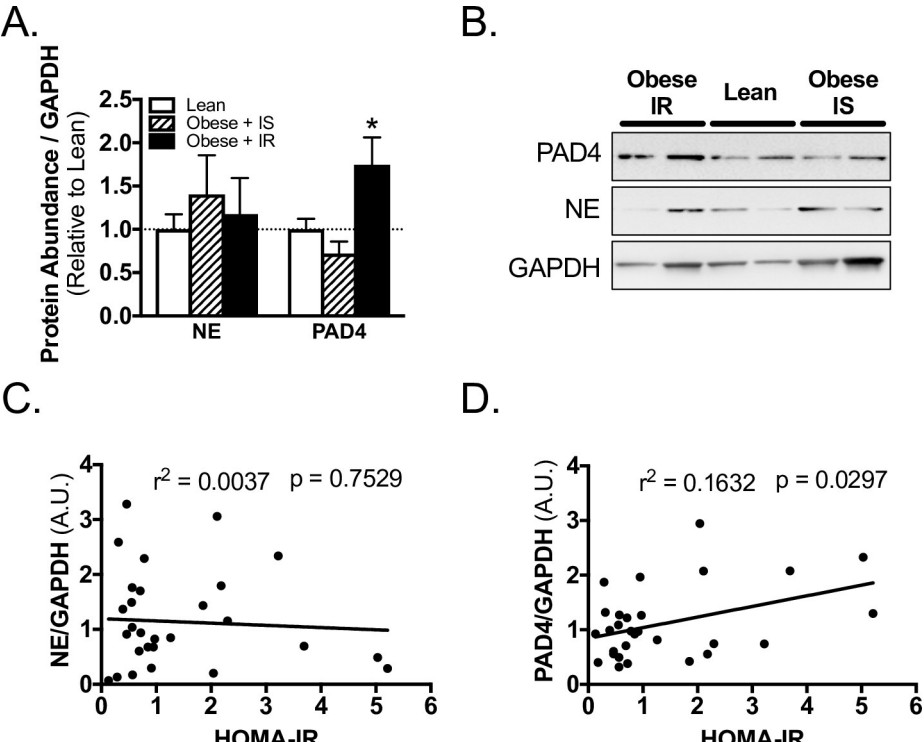

**Fig 3. PAD4 abundance correlates with insulin resistance in TJA subjects.** A) Western blot quantification and B) representative blots for PAD4, NE, and loading control GAPDH abundance in skeletal muscle of lean, obese insulin sensitive (IS), and obese insulin resistant (IR) subjects. Linear regression of C) NE, and D) PAD4 with HOMA-IR of subjects. *, p < 0.05 one-way ANOVA, Sidak's multiple comparison test versus lean. Data reported as mean ± SEM.

[3, 4], and insulin resistance and type 2 diabetes [12–15, 31] are significant risk factors for post-operative delayed wound healing in both rodents and humans. Remarkably, however, the mechanisms underlying this remains to be elucidated, which in turn makes it difficult to establish recommendations for pre- or post-surgical treatment of these patients. Considering the interrelationship between insulin resistance, NET formation in wounds and delayed wound healing [14, 16], herein we investigated the association between markers of NETosis and obesity and insulin resistance in total joint arthroplasty patients as a possible reason for delayed wound healing in these patients. Our results demonstrate that markers of NETosis in skeletal muscle correlate positively with insulin resistance in total joint arthroplasty patients, but not with obesity.

PAD4 is the initial and most vital protein needed for NET formation [20, 21, 32, 33]. In fact, in mouse and *in vitro* models, both chemical inhibition [18–20] and genetic ablation [21, 33] of PAD4 is sufficient to inhibit NET formation. Notably, the genetic ablation of PAD4 also improves re-epithelialization and the speed of wound healing in a dermal biopsy punch mouse model [14]. In the present study, we demonstrate that although PAD4 abundance in skeletal muscle does not correlate with obesity, it does correlate positively with insulin resistance. Moreover, the relationship between PAD4 abundance and insulin resistance is obesity independent. This is demonstrated in two ways. First, we find that obese insulin sensitive subjects have comparable PAD4 abundance to lean subjects. Second, when using multiple linear regression we find that PAD4 correlates positively with HOMA-IR, but does not correlate with BMI. Since NET formation and protein expression of PAD4 are increased in neutrophils from subjects with diabetes [14], PAD4 may be a valid biomarker for NET formation and delayed

wound healing in insulin resistant patients, suggesting that insulin resistance may serve as a better predictor of delayed wound healing than obesity.

NE and other proteases have long been implicated in delayed wound healing in humans via the degradation of fibronectin [34] and growth factors [35] which are needed for proper wound closure. Indeed, specialized cotton dressings have been engineered with the goal of removing NE and other proteinases from wound fluid to aid in wound closure [36]. We demonstrate that NE abundance in skeletal muscle tissue from the surgical site does not correlate with obesity or insulin resistance in total joint arthroplasty subjects. Although NE can be present in, and regulate the formation of, NETs [37], mice with a knockout of NE can still form NETs [32]. Thus, the presence of NE in the surgical site tissue may not correlate with NET formation, which may explain why PAD4 correlated with insulin resistance while NE did not.

Once neutrophils expel the NETs, NETs can be found in the plasma as cf-dsDNA [38, 39]. Interestingly, while NET formation is thought to be beneficial to wound health by sequestering and destroying pathogens [17, 22], at the same time it has been demonstrated to slow the process of wound healing [14, 15]. Indeed, breaking down NETs via treatment with DNase (as NETs are made of DNA) improves re-epithelialization and the speed of wound healing in a dermal biopsy punch mouse model [14]. In the present study, however, we observed no correlation between plasma cf-dsDNA and obesity or insulin resistance in total joint arthroplasty subjects. Notably, while the assessment of cf-dsDNA is objective and quantitative [40], it is not specific, and can be increased in a variety of conditions including cancer [38], systemic lupus erythemus [39] and rheumatoid arthritis [39], and even independent of NET formation [41]. Although we account for many of these conditions in our study's exclusion criteria, since cf-dsDNA can be present independent of NET formation whether there is any correlation between the measured NETs and HOMA-IR or BMI remains unanswered.

The findings of this study should be interpreted in the light of certain limitations. Firstly, our sample size was too low to thoroughly account for multiple co-morbidities and/or variables in our regression models and precluded us from making conclusions related to post-surgical wound healing. Indeed, rates of post-surgical complications due to delayed wound healing, namely surgical site infection, are very low especially for TJA and TKA [30], and therefore was not able to be statistically analyzed with the sample size of this preliminary report. Certainly, this will be of high interest in future prospective-focused studies to investigate whether markers of NETosis could be predictive of surgical risk in larger cohorts of patients. Another limitation of this study is the use of western blot analysis for assessing the relative abundance of PAD4 and NE in tissue. Western blot analysis is semi-quantitative in nature and therefore while it allows for comparisons and correlations to be made, it does not allow for measuring specific quantities of these proteins. Therefore, in future studies, it would be of interest to utilize more quantitative methods, such as ELISA, in order to quantify the precise concentrations of these proteins. Additionally, western blot analysis lacks the resolution to determine whether the PAD4 or NE levels that were measured were intrinsic or extracellular as part of the NET complex. Indeed, while PAD4 abundance in skeletal muscle correlated with insulin resistance, NE did not which may be due to the broad measuring of NE in tissue via western blot. The future use of immunostaining and/or sandwich ELISA assays would allow for the assessment of extracellular DNA co-localizing with PAD4, NE and/or myeloperoxidase as well as protein-protein complexes such as NE-myeloperoxidase, which would be more specific measures of NET formation. Furthermore, while the wound milieu is made up of various cell types (e.g. skin, adipose, muscle, immune), that all must be healthy in order to have proper wound healing, herein we chose to focus solely on skeletal muscle from the surgical site. Skeletal muscle undergoes early infiltration by neutrophils during acute injury [42, 43] (e.g. during surgery), and so it is a highly relevant tissue for studying NET formation in the context of

wound healing, however future studies may consider investigating other tissue types as well. An additional limitation is that granulocytes and macrophages also express PAD4 [44], not just neutrophils. However, we believe that our results can be interpreted as PAD4 of mainly neutrophil origin since eosinophils and basophils are not found in abundant numbers in muscle [45, 46], and macrophages only infiltrate muscle ~2 days post muscle injury [42]. Lastly, because samples were collected from subjects during surgery, reactive (i.e. due to a stimuli) NET formation could not be determined. Importantly, however, spontaneous NETosis is also increased in neutrophils from patients with diabetic ulcers [15], mice being fed a high fat diet [47], and neutrophils exposed to hyperglycemic conditions [16]. Thus, NETosis still occurs in diabetic and insulin resistant models without stimuli, and the results of this study should be interpreted in the context of spontaneous NET formation. Despite these limitations, a strength of this study is that, to our knowledge, it is the first study to evaluate markers of NETosis in obese and insulin resistant patients undergoing total joint arthroplasty.

## Conclusions

In conclusion, our data suggest that insulin resistant total joint arthroplasty subjects have a higher capacity for NET formation in tissue (i.e. skeletal muscle) in proximity of the surgical site, which could increase their risk for post-operative surgical site delayed wound healing. In future studies, in a larger cohort of subjects, it will be interesting to determine if surgical site PAD4 abundance is predictive of delayed wound healing or surgical site infection. Furthermore, it will be of interest to see if the trends found in the present study hold for other orthopedic-related procedures for which obesity and/or diabetes are associated with impaired healing and post-operative recovery and outcomes, such as shoulder, wrist, or ankle surgery. Finally, with NET formation contributing to delayed wound healing [14, 48] and diabetes priming neutrophils to undergo NETosis [14, 16], future research should investigate the impact of interventions or treatments that improve insulin sensitivity (e.g. exercise, dietary intervention) or reduce NETs at the surgical site (e.g. DNase treatment) on surgical outcomes in insulin resistant and/or obese patients. In the short-term, surgeons might consider accounting for insulin resistance, more so than obesity, as a pre-operative marker for the potential of post-surgical complications.

## Supporting information

**S1 Table. Subject NETosis markers—regression table.**
(XLSX)

**S1 Data.**
(XLSX)

## Author Contributions

**Conceptualization:** Vitor F. Martins, Christopher R. Dobson, Simon Schenk.

**Data curation:** Vitor F. Martins, Jesal Parekh, Jan M. Hughes-Austin.

**Formal analysis:** Vitor F. Martins, Christopher R. Dobson, Maedha Begur, Jan M. Hughes-Austin.

**Funding acquisition:** Christopher R. Dobson.

**Investigation:** Vitor F. Martins, Christopher R. Dobson, Maedha Begur.

**Methodology:** Jesal Parekh, Scott T. Ball, Francis Gonzalez, Jan M. Hughes-Austin.

Project administration: Vitor F. Martins, Christopher R. Dobson, Jesal Parekh, Simon Schenk.

Resources: Jesal Parekh, Scott T. Ball, Francis Gonzalez, Simon Schenk.

Supervision: Scott T. Ball, Francis Gonzalez, Simon Schenk.

Visualization: Simon Schenk.

Writing – original draft: Vitor F. Martins.

Writing – review & editing: Simon Schenk.

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
