## [Decision Letter · Decision Letter 0]

25 Aug 2020

PONE-D-20-15536

Surgical site peptidyl deaminase 4 (PAD4), a biomarker of NETosis, correlates with insulin resistance in total joint arthroplasty patients

PLOS ONE

Dear Dr. Schenk,

Thank you for submitting your manuscript to PLOS ONE. After careful consideration, we feel that it has merit but does not fully meet PLOS ONE’s publication criteria as it currently stands. Therefore, we invite you to submit a revised version of the manuscript that addresses the points raised during the review process.

Consequently, after considering the commentaries made by three reviewers, Major Revision is requested.

As you may appreciate, the reviewers agree in their concerns regarding certain issues related to sample size, methodological approach, and statistical analysis.

Likewise, such a decision is justified by taking into account the publication requirements for PLOS ONE regarding that the papers must be technically sound and the data should support the conclusions.

Therefore, please carefully address each of the points requested in the comments provided by the reviewers (please, see below), which can help to improve and strengthen your submission.

We look forward to receiving your revised manuscript.

Kind regards,

Maria Carmen Iglesias-Osma, M.D., Ph.D.

Academic Editor

PLOS ONE

Journal Requirements:

2.This study received Institutional Review Board approval (IRB: 151506; Approved 10/22/2015) and written informed consent was obtained from each patient before the initiation of the study.'

(a) Please amend your current ethics statement to include the full name of the ethics committee/institutional review board(s) that approved your specific study.

(b) Once you have amended this/these statement(s) in the Methods section of the manuscript, please add the same text to the “Ethics Statement” field of the submission form (via “Edit Submission”).

3. Please provide a sample size and power calculation in the Methods, or discuss the reasons for not performing one before study initiation.

4.We note that you have indicated that data from this study are available upon request. PLOS only allows data to be available upon request if there are legal or ethical restrictions on sharing data publicly. For information on unacceptable data access restrictions, please see http://journals.plos.org/plosone/s/data-availability#loc-unacceptable-data-access-restrictions.

Reviewers' comments:

Reviewer's Responses to Questions

**Comments to the Author**

1. Is the manuscript technically sound, and do the data support the conclusions?

Reviewer #1: Partly

Reviewer #2: No

Reviewer #3: No

2. Has the statistical analysis been performed appropriately and rigorously? 

Reviewer #1: No

Reviewer #2: Yes

Reviewer #3: Yes

3. Have the authors made all data underlying the findings in their manuscript fully available?

Reviewer #1: No

Reviewer #2: No

Reviewer #3: Yes

4. Is the manuscript presented in an intelligible fashion and written in standard English?

Reviewer #1: Yes

Reviewer #2: Yes

Reviewer #3: Yes

5. Review Comments to the Author

Reviewer #1: In their manuscript titled “Surgical Site Peptidyl Deaminase 4 (PAD4), a Biomarker of NETosis, Correlates with Insulin Resistance in Total Joint Arthroplasty Patients”, Martins and colleagues measured biomarkers of NETosis in lean and obese TJA (Total Joint Arthroplasty) patients. Among the biomarkers of NETosis, only the surgical site PAD4 level was found significantly elevated in insulin resistant obese subjects, but not in insulin sensitive obese subjects, compared to lean subjects. PAD4 level also showed significant positive correlation with HOMA IR but not with BMI. This information appears new and important. The manuscript is well written and easy to read and understand. However following issues need to be critically considered:

1. The sample size was too low and the PAD4 was measured by immunoblotting, a semi-quantitative method. Therefore, the findings of correlation and regression analyses are not much convincing. More quantitative ELISA or activity assay of PAD4 would have been appropriate in this setting.

2. In the Method section, it is not clear how the skeletal muscle samples were homogenized. It is also not clear whether the total tissue extract or the nuclear fractions were collected for immunoblotting. Method of quantification of the blots was not mentioned.

3. In the statistical method section, the regression models were not clearly described.

4. In Table 3 (Supplementary Table 1), only the ‘beta’ values are shown and the actual ‘p’values are not shown. The ‘beta’ value is only about 0.2 for the dependent variable PAD4 with HOMA-IR, which is not so impressive.

5. Age in Table 1 and age, weight and BMI in Table 2 appear to be significantly different between the two groups. Please recheck the statistical analyses; and if there is any mistake please rewrite the manuscript accordingly.

Reviewer #2: Neutrophil extracellular traps (NETs) have been previously shown to delay wound healing, in particular in diabetes. The authors aimed to elucidate the relationship between NETs, obesity and insulin resistance in total joint arthroplasty. While the objective of the study is sound, there are a number of concerns in methodologies and data interpretation.

1. Figure 1. Cell-free double-stranded DNA (cf-dsDNA) is no longer an acceptable assay for NETs, as it also detects non-NET extracellular DNA from other cell deaths (like authors mentioned in the discussion). Myeloperoxidase (MPO)-DNA complex or neutrophil elastase (NE)-DNA complex ELISA should be performed. Whether there are any correlations between NETs versus BMI or HOMA-IR remains unanswered.

2. Figure 2 & 3. While increased PAD4 expression in neutrophils may explain an increased propensity for neutrophils to undergo NETosis (Wong et al., 2015), tissue PAD4 level per se is NOT a biomarker of NETs / NETosis. PAD4 is constitutively expressed in neutrophils; therefore, an increase in PAD4 level in tissue may simply mean that there are more neutrophils recruited to the inflamed tissue. Western blot analysis lacks a resolution of whether PAD4 is extracellularly released in NETosis and whether it is coated on NETs. In fact, a recent study also showed a possibility that PAD4 can be expressed on cell surface independent of NETosis (Zhou et al., 2017). Therefore, determining tissue NET levels by blotting for total PAD4 and NE did not provide support to the authors’ conclusion. Immunostaining of tissues (externalized DNA co-localizing with NE, MPO or citrullinated histone H3) is essential to prove for the presence of NETs.

3. Figure 3. As for data interpretation, why NE did not go in the same trend as PAD4 needs further experimental justification. Although NETosis can occur independent of NE (Martinod et al., 2016), extracellular NE co-localizing with DNA (immunostaining) and NE-MPO complex (ELISA) are standard definition of NETs: In the end, the enzyme is expelled along with the decondensed chromatin and coated on the DNA. If NETosis does occur, the level of extracellular DNA-associated NE will increase regardless whether NE plays a role in NETosis. The current data seem to imply that NETosis may not be involved.

4. The significance of the study seems rather weak. How do the findings impact on patient care / treatment? Is there any correlation between NET levels in the surgical site (peri-operation) and post-surgical / recovery outcomes in these patients? Whether there is a causal relationship between peri-operation NET level and post-surgical outcome should also be explored.

5. Concept clarification – Higher NET content in surgical site may not necessarily link to higher risk of infection (comment on the last sentence in abstract). How NETs interact with different bacterial species in wounds is actually an open question; such generalization is therefore not justified. In fact, this notion also contradicts the introduction where authors indicate “… NET formation would be beneficial to wound health by sequestering and destroying pathogens”.

6. PAD4 is spelt wrongly throughout the manuscript. It is “peptidylarginine deiminase 4”.

7. References are duplicated (e.g. #15 and #23 are the same).

Reviewer #3: The work is interesting, however,

1) As the authors mention, the main problem is the number of cases, which when divided between obese and lean, insulin resistant and insulin sensitive, finally is an n of 3 and 4, which cannot be considered significant.

2) They were able to do a postoperative follow-up to the patients to see their evolution and compare them with PAD4, if they already had these data.

3) Were knee surgeries performed with ischemia or without ischemia? This is an important piece of information to see the evolution of the patient and could also be a piece of information worth taking into account, since with ischemia / reperfusion there is a greater oxidative stress and therefore inflammation and greater ease for the formation of NETs.

4) In case the other reviewers and / or the Editor decide to accept it for publication, the title should say: "preliminary report"

6. PLOS authors have the option to publish the peer review history of their article (what does this mean?). If published, this will include your full peer review and any attached files.

Reviewer #1: No

Reviewer #2: No

Reviewer #3: No

---

## [Author Response · Author response to Decision Letter 0]

30 Dec 2020

Journal Requirements

1. Please ensure that your manuscript meets PLOS ONE's style requirements

This has been updated.

2. This study received Institutional Review Board approval (IRB: 151506; Approved 10/22/2015) and written informed consent was obtained from each patient before the initiation of the study. (a) Please amend your current ethics statement to include the full name of the ethics committee/institutional review board(s) that approved your specific study. (b) Once you have amended this/these statement(s) in the Methods section of the manuscript, please add the same text to the “Ethics Statement” field of the submission form (via “Edit Submission”).

This has now been added to the methods (Lines 75-76) and has been updated within the “Ethics Statement” field of the submission form.

3. Please provide a sample size and power calculation in the Methods, or discuss the reasons for not performing one before study initiation.

This has now been added to the methods (Lines 134-136).

4. We note that you have indicated that data from this study are available upon request. PLOS only allows data to be available upon request if there are legal or ethical restrictions on sharing data publicly. For information on unacceptable data access restrictions, please see http://journals.plos.org/plosone/s/data-availability#loc-unacceptable-data-access-restrictions. In your revised cover letter, please address the following prompts: a) If there are ethical or legal restrictions on sharing a de-identified data set, please explain them in detail (e.g., data contain potentially identifying or sensitive patient information) and who has imposed them (e.g., an ethics committee). Please also provide contact information for a data access committee, ethics committee, or other institutional body to which data requests may be sent. b) If there are no restrictions, please upload the minimal anonymized data set necessary to replicate your study findings as either Supporting Information files or to a stable, public repository and provide us with the relevant URLs, DOIs, or accession numbers. Please see http://www.bmj.com/content/340/bmj.c181.long for guidelines on how to de-identify and prepare clinical data for publication. For a list of acceptable repositories, please see http://journals.plos.org/plosone/s/data-availability#loc-recommended-repositories.

The availability of the data has been updated to “no restriction” and a minimal anonymized data set has been uploaded as a supporting information file. 

Reviewer 1

We thank Reviewer 1 for their careful and detailed reviewing of our manuscript and for noting that aspects of our manuscript “appears new and important” and that our “manuscript is well written and easy to read and understand”. 

1. The sample size was too low and the PAD4 was measured by immunoblotting, a semi-quantitative method. Therefore, the findings of correlation and regression analyses are not much convincing. More quantitative ELISA or activity assay of PAD4 would have been appropriate in this setting.

To these excellent points, while we did not find significant differences in neutrophil elastase or cell-free DNA among groups, we now provide sample size calculations (based on sample size calculations utilizing effect sizes from previously published data; Methods, Lines 134-136), which demonstrate that our study was powered to detect differences in our measured parameters. Moreover, in the Discussion we have noted the methodological limitations of western blotting and included a statement to the Reviewer’s point that an ELISA approach would be beneficial for future work (Discussion, Lines 261 – 262). It is also important to note, that while we did not use the ELISA approach in this study, the western blot approach can still provide important and relevant data in terms of correlation and regression analysis.

2. In the Method section, it is not clear how the skeletal muscle samples were homogenized. It is also not clear whether the total tissue extract or the nuclear fractions were collected for immunoblotting. Method of quantification of the blots was not mentioned.

The method of homogenization for skeletal muscle (Lines 116 – 121), use of total tissue extract (Line 123), and method for quantification of immunoblots (Lines 130 – 131) have now been clarified in the methods.

3. In the statistical method section, the regression models were not clearly described.

This has been updated within the methods (Lines 141 – 143). 

4. In Table 3 (Supplementary Table 1), only the ‘beta’ values are shown and the actual ‘p’values are not shown. The ‘beta’ value is only about 0.2 for the dependent variable PAD4 with HOMA-IR, which is not so impressive.

Supplementary table 1 has now been updated to show beta values, r2, F statistic, and p values for all regressions. Notably, regarding the beta value for PAD4, since PAD4 and NE are expressed as arbitrary units from western blot quantification, it is difficult to interpret the relevance of the beta or slope value for these variables and a more important statistic is the p value, as it demonstrates if the slope is significantly different from zero.

5. Age in Table 1 and age, weight and BMI in Table 2 appear to be significantly different between the two groups. Please recheck the statistical analyses; and if there is any mistake please rewrite the manuscript accordingly.

To this impotant point, we have re-evaluated all statistical analyses and they are correctly described in the manuscript.

Reviewer 2

We thank Reviewer 2 for their insightful reviewing of our manuscript and for noting that “the objective of the study is sound”.

1. Figure 1. Cell-free double-stranded DNA (cf-dsDNA) is no longer an acceptable assay for NETs, as it also detects non-NET extracellular DNA from other cell deaths (like authors mentioned in the discussion). Myeloperoxidase (MPO)-DNA complex or neutrophil elastase (NE)-DNA complex ELISA should be performed. Whether there are any correlations between NETs versus BMI or HOMA-IR remains unanswered.

While we tried to limit the impact of this potentially confounding variable by having stringent exclusion criteria, this concern cannot be completely controlled. Furthermore, cf-dsDNA can be present in serum independent of NET formation. To the Reviewer’s point, we have expanded the part of the Discussion that originally addressed this limitation, and now include additional discussion on the use of MPO-DNA and NE-DNA complex ELISA methods (See Discussion, Lines 262 – 269).

2. Figure 2 & 3. While increased PAD4 expression in neutrophils may explain an increased propensity for neutrophils to undergo NETosis (Wong et al., 2015), tissue PAD4 level per se is NOT a biomarker of NETs / NETosis. PAD4 is constitutively expressed in neutrophils; therefore, an increase in PAD4 level in tissue may simply mean that there are more neutrophils recruited to the inflamed tissue. Western blot analysis lacks a resolution of whether PAD4 is extracellularly released in NETosis and whether it is coated on NETs. In fact, a recent study also showed a possibility that PAD4 can be expressed on cell surface independent of NETosis (Zhou et al., 2017). Therefore, determining tissue NET levels by blotting for total PAD4 and NE did not provide support to the authors’ conclusion. Immunostaining of tissues (externalized DNA co-localizing with NE, MPO or citrullinated histone H3) is essential to prove for the presence of NETs.

This is an excellent point. As PAD4 is constitutively expressed in neutrophils, it is possible that the increased levels of PAD4 seen are secondary to increased neutrophil content in the muscle due to an inflammatory state and not necessarily increased NET formation. We attempted to minimize this confounding variable by only including subjects with degenerative osteoarthritis as their indication for surgery, thus the inflammatory state of the muscle should be minor, or at least normalized throughout subjects. Furthermore, we excluded patients from this study that had conditions that may cause an increased inflammatory state including malignancy, inflammatory arthritis, infection, and trauma. 

With normalizing the potential for neutrophil infiltration into the tissue, PAD4 expression via western blot, as others have assessed (PMIDs: 26076037, 33081303), can provide some insight into NET formation. However, western blot analysis does lack the resolution to assess whether PAD4 (or NE) is extracellular as part of the NET complex, or intracellular/cell surface attached. Thus, we now include further discussion of the limitations of western blotting in assessing NETosis and include the excellent suggestions from the Reviewers as potential future experiments (See Discussion, 257-269).

3. Figure 3. As for data interpretation, why NE did not go in the same trend as PAD4 needs further experimental justification. Although NETosis can occur independent of NE (Martinod et al., 2016), extracellular NE co-localizing with DNA (immunostaining) and NE-MPO complex (ELISA) are standard definition of NETs: In the end, the enzyme is expelled along with the decondensed chromatin and coated on the DNA. If NETosis does occur, the level of extracellular DNA-associated NE will increase regardless whether NE plays a role in NETosis. The current data seem to imply that NETosis may not be involved.

As discussed in point #2, while western blotting can provide some insight into NET formation, it lacks the resolution to assess whether NE is extracellular as part of the NET complex, or intracellular/cell surface attached, which can explain the lack of difference in NE levels. Thus, we now include further discussion of the limitations of western blotting in assessing NETosis and include the excellent suggestions from the Reviewers of utilizing immuinostaining and sandwich ELISA assays as potential future experiments (See Discussion, 257-269).

4. The significance of the study seems rather weak. How do the findings impact on patient care / treatment? Is there any correlation between NET levels in the surgical site (peri-operation) and post-surgical / recovery outcomes in these patients? Whether there is a causal relationship between peri-operation NET level and post-surgical outcome should also be explored.

We were able to do post-operative analysis to assess for surgical site infections within one month of the surgery. However, due to the rarity of surgical site infection in this surgery type (we only had two documented cases), we were not able to do statistical analysis on this data. We have now included this information within the manuscript (Lines 252 – 255). To the Reviewer’s point, based on this work we are now looking to conduct prospective-focused studies to better investigate whether markers of NETosis could be predictive of surgical risk and recovery.

5. Concept clarification – Higher NET content in surgical site may not necessarily link to higher risk of infection (comment on the last sentence in abstract). How NETs interact with different bacterial species in wounds is actually an open question; such generalization is therefore not justified. In fact, this notion also contradicts the introduction where authors indicate “… NET formation would be beneficial to wound health by sequestering and destroying pathogens”.

The Reviewer raises a highly pertinent point as to whether NET formation is being beneficial or a hinderance to wound health. As we explore in the manuscript, NETs can sequester and destroy pathogens however NET formation also slows wound healing, both of which are at odds with each other in terms of overall wound health and the potential for surgical complications. We now include additional text in Abstract (Line 32) and Introduction (Lines 59 – 62) that further discusses and clarifies this point as being an open question in the field. 

6. PAD4 is spelt wrongly throughout the manuscript. It is “peptidylarginine deiminase 4”.

This has been corrected throughout the manuscript.

7. References are duplicated (e.g. #15 and #23 are the same).

This has been corrected and references have been checked for similar duplications.

Reviewer 3

1. As the authors mention, the main problem is the number of cases, which when divided between obese and lean, insulin resistant and insulin sensitive, finally is an n of 3 and 4, which cannot be considered significant.

This study has a total of 29 subjects, 15 of which were classified as lean, and 14 of which were classified as obese (See, Table 1). Within the 14 obese subjects, 7 were insulin sensitive and 7 were insulin resistant (See, Table 2). The end n for each group: Lean = 15; Obese insulin sensitive = 7; Obese insulin resistant = 7. We now make this clearer in the Methods. 

2. They were able to do a postoperative follow-up to the patients to see their evolution and compare them with PAD4, if they already had these data.

We were able to do post-operative analysis to assess for surgical site infections within one month of the surgery. However, due to the rarity of surgical site infection in this surgery type (we only had two documented cases), we were not able to do statistical analysis on this data. We have now included this information within the manuscript (Lines 252 – 255). To the Reviewer’s point, based on this work we are now looking to conduct prospective-focused studies to better investigate whether markers of NETosis could be predictive of surgical risk and recovery.

3. Were knee surgeries performed with ischemia or without ischemia? This is an important piece of information to see the evolution of the patient and could also be a piece of information worth taking into account, since with ischemia / reperfusion there is a greater oxidative stress and therefore inflammation and greater ease for the formation of NETs.

To this excellent point, sample collection for both hip and knee surgeries were performed without ischemia. We have updated the Methods to include this point. 

4. In case the other reviewers and / or the Editor decide to accept it for publication, the title should say: "preliminary report"

This has been updated in the title.

---

## [Editor Report · Decision Letter 1]

5 Jan 2021

Surgical site peptidylarginine deaminase 4 (PAD4), a biomarker of NETosis, correlates with insulin resistance in total joint arthroplasty patients: A preliminary report

PONE-D-20-15536R1

Dear Dr. Schenk,

We’re pleased to inform you that your manuscript has been judged scientifically suitable for publication and will be formally accepted for publication once it meets all outstanding technical requirements.

Kind regards,

Maria Carmen Iglesias-Osma, M.D., Ph.D.

Academic Editor

PLOS ONE

---

## [Editor Report · Acceptance letter]

11 Jan 2021

PONE-D-20-15536R1 

Surgical site peptidylarginine deaminase 4 (PAD4), a biomarker of NETosis, correlates with insulin resistance in total joint arthroplasty patients: A preliminary report 

Dear Dr. Schenk:

I'm pleased to inform you that your manuscript has been deemed suitable for publication in PLOS ONE. Congratulations! Your manuscript is now with our production department. 

Kind regards, 

on behalf of

Prof. Dr. Maria Carmen Iglesias-Osma 

Academic Editor

PLOS ONE